# A Novel Semi-Solid Self-Emulsifying Formulation of Aprepitant for Oral Delivery: An In Vitro Evaluation

**DOI:** 10.3390/pharmaceutics15051509

**Published:** 2023-05-16

**Authors:** Hakan Nazlı, Burcu Mesut, Özlem Akbal-Dağıstan, Yıldız Özsoy

**Affiliations:** 1Department of Pharmaceutical Technology, Trakya University, 22030 Edirne, Turkey; hakannazli@trakya.edu.tr; 2Department of Pharmaceutical Technology, Istanbul University, 34116 Istanbul, Turkey; bmesut@istanbul.edu.tr (B.M.); ozlemakbal@istanbul.edu.tr (Ö.A.-D.)

**Keywords:** supersaturated SNEDDS, capsule filled SNEDDS, semi-solid SNEDDS, solubility enhancement, aprepitant

## Abstract

Aprepitant is the first member of a relatively new antiemetic drug class called NK_1_ receptor antagonists. It is commonly prescribed to prevent chemotherapy-induced nausea and vomiting. Although it is included in many treatment guidelines, its poor solubility causes bioavailability issues. A particle size reduction technique was used in the commercial formulation to overcome low bioavailability. Production with this method consists of many successive steps that cause the cost of the drug to increase. This study aims to develop an alternative, cost-effective formulation to the existing nanocrystal form. We designed a self-emulsifying formulation that can be filled into capsules in a melted state and then solidified at room temperature. Solidification was achieved by using surfactants with a melting temperature above room temperature. Various polymers have also been tested to maintain the supersaturated state of the drug. The optimized formulation consists of Capryol^TM^ 90, Kolliphor^®^ CS20, Transcutol^®^ P, and Soluplus^®^; it was characterized by DLS, FTIR, DSC, and XRPD techniques. A lipolysis test was conducted to predict the digestion performance of formulations in the gastrointestinal system. Dissolution studies showed an increased dissolution rate of the drug. Finally, the cytotoxicity of the formulation was tested in the Caco-2 cell line. According to the results, a formulation with improved solubility and low toxicity was obtained.

## 1. Introduction

Nausea and vomiting are among the most irritating side effects of chemotherapy. About 75% of cancer patients suffer from chemotherapy-induced nausea and vomiting (CINV) [1]. CINV should be avoided as it may delay chemotherapy or cause patients to refuse treatment completely [2]. Therefore, antiemetic treatment guidelines have been prepared by different organizations related to cancer treatment such as the Multinational Association for Supportive Care in Cancer (MASCC), European Society for Medical Oncology (ESMO), American Society of Clinical Oncology (ASCO), National Comprehensive Cancer Network (NCCN) [3]. Prophylaxis has been accepted as the first goal of antiemetic therapy by all leading guidelines. As a basic principle, antiemetic prophylaxis should be applied to all patients with a risk of CINV above 10% [4].

There are various pathways triggering chemotherapy-induced nausea and vomiting. Although the mechanism has not been clearly defined yet, researches show that CINV may occur mainly through neurotransmitters (serotonin, dopamine, substance P, acetylcholine) in the gastrointestinal tract and central nervous system [5]. Chemotherapeutic drugs can initiate the emetic reflex by activating neurotransmitter receptors located in the area postrema region of the brain or at the terminal ends of vagal neurons close to enterochromaffin cells in the GI tract. Both conditions send impulses to organs and tissues to cause vomiting [6].

The main approach to control CINV is to block neurotransmitters and their receptors which mediate the generation of impulses from sensory neurons in the GI tract (5-HT_3_, D_2_, NK_1_) and central nervous system (5-HT_3_, D_2_, NK_1_, ACh_1_, opioid μ). Therapeutic agents that can block neurotransmitter receptors in the vomiting center, chemoreceptor trigger zone (CTZ), and GI tract are useful in preventing or controlling CINV [5,7]. Antiemetic prophylaxis is usually performed with a combination of drugs. Dexamethasone, serotonin, and neurokinin-1 (NK_1_) receptor antagonists are drugs that can be used for this purpose [8].

NK_1_ receptor antagonists are the newest class of drugs introduced in the early 2000s for the treatment of emesis [6]. Aprepitant (APR) is the first commercial drug from the NK_1_ receptor antagonists. Its mechanism of action is decreasing the emetic effects of substance P, which is a neurokinin in the central nervous system [9]. Its physical appearance is a white to off-white, crystalline solid powder with a molecular weight of 534.4 g/mol. APR is a slightly alkaline compound with a pK_a_ of 9.7. It has a relatively high logP (about 4.8) and melting point (254 °C) [10,11]. There is no consensus in the literature on the BCS status of APR. It can be categorized as BCS Class II [12] or Class IV [13] molecule, which is most likely due to the permeability value being near the limit of Class II and Class IV.

The main problem with this drug is its poor solubility. It is practically insoluble over the wide pH range in water [14]. Low solubility in water is one of the most important physical disadvantages which restrict the usage of new chemical entities as drug compounds. Drugs with low solubility in the gastrointestinal system slow drug release and cause incomplete absorption [15]. Particle size reduction was used in the commercial formulation of APR (Emend^®^) to provide an effective treatment [16]. NanoCrystal^®^ technology developed by Elan Corporation is used in the nano-sized formulation of APR. Drug particles are milled to submicron size (approximately 0.12 µm) and stabilized with polymer and surfactant [17]. Production with this method consists of many successive steps that cause the cost of the drug to increase. Therefore, despite the improved bioavailability of Emend^®^, efforts to enhance the dissolution rate of APR have continued since it was first introduced [18].

Various formulations based on nano-sized APR such as solid dispersions [19], microemulsions [20], cyclodextrin complexes [21], orally disintegrating film [22], solid solution [23], and surfactant-based formulations [15] have been reported in the literature. Previously, our research group also developed a solidified self-nanoemulsifying drug delivery system (SNEDDS) of APR using porous adsorbents [24].

SNEDDS are homogenous liquid mixtures that form o/w-type nanoemulsions spontaneously with droplet sizes typically below 200 nm in an aqueous media. Liquid SNEDDS can be converted to solid dosage forms with different approaches [25]. SNEDDS have many advantages discussed elsewhere in increasing the bioavailability of poorly soluble drugs [26]. However, like other drug delivery systems, SNEDDS have some disadvantages. One of the potential drawbacks is their limited usability with active pharmaceutical ingredients (API) used in relatively high doses. SNEDDS are more applicable to potent drugs otherwise; a large amount of SNEDDS preconcentrate will be required to dissolve the API [27]. A high amount of SNEDDS preconcentrate increases the amount of adsorbent used when converting the formulation into a solid dosage form. Consequently, the volume of powder for a single dosage form increases. The high amount of powder may restrict capsule filling, and tablet compression due to the limited volume of capsule shells or tablet dies. In addition, patient compliance may also be adversely affected. Our previously developed formulation containing 40 mg of APR consists of 1.14 g of liquid SNEDDS and 0.76 g of adsorbent. The powder amount, which reaches 1.9 g (approx. 5 mL) in total, suffers from the above-mentioned problem. In this study, we aimed to develop solid APR-loaded SNEDDS formulations with a different solidification technique without using a high amount of adsorbent. Briefly, we designed a SNEDDS formulation using a surfactant with a melting point above room temperature. Thus, a self-solidifying SNEDDS formulation was obtained after the warm melt was filled into the capsule shell.

## 2. Materials and Methods

### 2.1. Materials

APR was generously gifted by Platin Kimya (Istanbul, Turkey). Imwitor^®^ 988 was kindly gifted by IOI Oleochem. Kolliphor^®^ CS20, Kolliphor^®^ P188, Cremophor^®^ A25, and Soluplus^®^ were generously gifted by BASF (Ludwigshafen, Germany). Transcutol^®^ P, Capryol™ 90, Gelucire^®^ 44/14, and Gelucire^®^ 48/16 were generously gifted by Gattefossé (Lyon, France). Methanol (LiChrosolv^®^ grade) was purchased from Merck Millipore (Darmstadt, Germany). Bile salt (B-3883), pancreatic extract (P-1625), and 4-bromobenzeneboronic acid (B-75956), Trizma^®^ maleate (T-3128), calcium chloride dihydrate (223505), sodium chloride (S9888), and phosphoric acid (695017) were purchased from Sigma-Aldrich (Darmstadt, Germany). Lipoid S-100 was purchased from Lipoid GmbH (Ludwigshafen, Germany). All the other chemicals used were analytical grade.

### 2.2. Method for Quantification of APR

Thermo Surveyor HPLC system (Temecula, CA, USA) consisting of a pump, an autosampler, and a UV–Vis detector was used for APR quantification. Chromatographic analysis was performed on Waters^TM^ Symmetry (4.6 mm × 250 mm 5 µm) C18 analytical column. The mobile phase was a mixture of methanol and pH 3 phosphoric acid solution (80:20 *v*/*v*). Mobile phase content and flow rate (0.8 mL/min) was constant throughout the analysis period (10 min). The temperature of HPLC system was set to 25 °C. A volume of 25 µL sample was injected into the column, and the APR was detected at 210 nm. The concentrations of the APR standards were prepared between 1 and 20 µg/mL to obtain the calibration curve. Method was validated according to the ICH Q2 (R1) guideline. All excipients used in this study did not interfere with the assay of APR.

### 2.3. Screening of Surfactants

The nano-emulsification ability of surfactants was screened by a spectrophotometric method determined by Date and Nagarsenker [28]. Oil and surfactant were weighed in equal amounts. The mixture was heated at 60 °C, for the homogenization of the components. Then, the mixture was diluted 100-fold with distilled water to obtain nanoemulsion. The appearance of the dispersion formed was evaluated by % transmittance measurement at 638 nm with UV-1601 (Shimadzu, Kyoto, Japan) spectrophotometer. Distilled water was used as blank.

### 2.4. Equilibrium Solubility of APR

The solubility of APR in surfactant solutions (1%) was determined using the shake flask method. In brief, an excess amount of APR was mixed with the test medium in a test tube to obtain a saturated solution. Tube was vortexed (Daihan, Republic of Korea) for 60 s to homogeneously disperse APR crystals in the liquid. Mixtures were shaken in a water bath (100 rpm, 37 °C) for 24 h. Then, the tubes were centrifuged at 12,225× *g* for 15 min. Drug concentration in the supernatant was obtained via the HPLC method.

### 2.5. Nanoemulsion Droplet Size Measurements

Prepared SNEDDS formulations was diluted 100-fold with distilled water. The droplet size and polydispersity index (PDI) measurements of the formed nanoemulsion were carried out in 173° backscattering mode using Malvern Zetasizer Nano ZS (Malvern, UK) based on the dynamic light scattering (DLS) technique. The sample was filled into disposable polystyrene cuvettes, and the measurements were carried out at 25 °C. Since the external phase of dispersion is water, distilled water was chosen as the dispersion medium (refractive index: 1.330 and viscosity: 0.8872). The dispersions showing a unimodal distribution, with a PDI value below 0.25 and a Z-average below 100 nm, were considered successful. All studies were repeated three times.

### 2.6. Short-Term Stability of Drug-Loaded Formulations

APR was loaded on the formulations that passed the preliminary quality test based on droplet size. Short-term stabilities of drug-loaded formulations (20–40 mg) were examined. Measurements were repeated at 0, 1, 2, and 4 h, and changes in the droplet size and PDI were analyzed to evaluate the short-term stability of the formed nanoemulsions.

### 2.7. Preparation of Solid SNEDDS Formulations

SNEDDS formulations were prepared by mixing all the selected excipients until the mixtures became transparent with the help of mild (60 °C) heat. Supersaturated SNEDDS were prepared by adding various polymeric precipitation inhibitors (PPI) into the formulations. Polymers (5% or 10% *w*/*w*) such as Soluplus^®^, PVP/VA (Kollidon^®^ VA64), PVP (Kollidon^®^ 25), and HPMC (Methocel^TM^ E5) were tested as PPIs. Finally, APR was added to the formulations and completely dissolved by mixing at 60 °C. Solid dosage forms were prepared by filling 00 gelatin capsule bodies while the mixture was still in the melt state [29].

### 2.8. In Vitro APR Precipitation

The precipitation of the drug from the nanoemulsion was determined with a method described by Pouton and Porter [30]. Briefly, SNEDDS preconcentrates diluted 100-fold in distilled water and obtained nanoemulsions were kept in a shaking water bath (37 °C and 100 rpm) during the test. Then, 0.5 mL samples were taken without volume replacement at predefined time intervals. The samples were immediately passed through a 0.45 µm filter. The filtered sample was immediately diluted in the mobile phase to prevent further precipitation. The APR concentration in filtrates was determined by HPLC. The cumulative (for 4 h) dissolved APR in formulations with polymers, which was compared to the polymer-free formulation by estimating the area under the curve (AUC), suggested by Quan et al. [31]. A higher AUC value indicated that the APR remained solubilized for a longer time without precipitate.

### 2.9. Characterization of Optimized SNEDDS Formulation

A series of characterization studies were carried out on the optimized formulation.

#### 2.9.1. Nature of the Dispersion Formed

The nature of the dispersion obtained was investigated by changing the mixing order of the excipients. Blank SNEDDS formulation was prepared by mixing Capryol^TM^ 90, Kolliphor^®^ CS20, and Transcutol^®^ P at 60 °C. The mixture was dispersed in water to form the nanoemulsion. As an alternative method, hydrophilic excipients (Kolliphor^®^ CS20 and Transcutol^®^ P) were dissolved in distilled water first, and then Capryol^TM^ 90 was added to this aqueous phase. The dispersions at 1/100 dilution prepared by that two methods were evaluated using the DLS method [32].

#### 2.9.2. % Transmittance Measurements

The % transmittance measurements of the formulations were carried out in distilled water and buffer solutions (pH 1.2, pH 4.5, and pH 6.8). SNEDDS was diluted 100-fold in the relevant medium, and the % transmittance values of the resulting dispersions were measured at a wavelength of 638 nm. Formulations that produced clear or slightly bluish dispersions in different media were considered to pass this test.

#### 2.9.3. pH Measurement of Dispersions

The pH values of the dispersions formed after the formulations were diluted 100 times with distilled water and measured by Mettler Toledo pH/Ion S220 pH meter at 37 °C.

#### 2.9.4. Cloud Point Measurement

The formulations were dispersed in water (100-fold) and then heated gradually. The dispersion was inspected visually, and the cloud point was determined by measuring the temperature at which a cloudy appearance occurred.

#### 2.9.5. Stress Tests

SNEDDS formulations were subjected to challenging conditions such as centrifugation, heating–cooling, and freeze–thawing cycles. Physical appearance was examined at the end of each stage. Formulations diluted 100-fold with distilled water centrifuged at 4000 rpm for 15 min. Six heating–cooling (40 °C/4 °C) and freeze–thaw (−20 °C/25 °C) cycles were separately performed for 48 h at each temperature in diluted dispersions. Then, the dispersions were checked for instability as phase separation or precipitation.

#### 2.9.6. Emulsification Efficiency and Self-Emulsification Time

Emulsification efficiency was determined using standard dissolution apparatus II. One gram of the formulation was added to 500 mL of distilled water at 37 °C, and the medium was stirred continuously at 50 rpm. The evaluation system described in the literature [33] was used to interpret the emulsification efficiency and duration.

#### 2.9.7. Robustness to Dilution

The selected SNEDDS formulations were evaluated for robustness upon 10-, 50-, 100-, 500-, and 1000-fold dilution in water, pH 1.2, pH 4.5, and pH 6.8 buffers. After dilution, the droplet size and distribution of the dispersions formed were measured. Zeta (ζ) potential measurements were performed in folded capillary cells and were calculated by the Zetasizer 7.13 software using the Helmholtz–Smoluchowski equation. Water (refractive index: 1.330 and viscosity: 0.8872, dielectric coefficient: 78.5) was chosen as the dispersion medium.

#### 2.9.8. Fourier Transform Infrared (FTIR) Spectroscopy

FTIR spectrum of samples was recorded using an IR spectrophotometer (Perkin Elmer Spectrum Two, MA, USA) in ATR mode equipped with a zinc selenide crystal. The scanning resolution was set to 1 cm^−1^ and the scan range was between 400 and 4000 cm^−1^. Each sample was measured in triplicate.

#### 2.9.9. Differential Scanning Calorimetry (DSC) Analysis

DSC analysis was performed with a TA Instruments DSC 250 instrument (New Castle, DE, USA). A 5 mg sample was weighed and sealed in hermetic aluminum pans (Tzero Hermetic Pan and Lid). A hermetically sealed empty aluminum pan was used as a reference. The samples were heated at a rate of 10 °C/minute between 30 and 350 °C. Nitrogen gas was used as the inert gas at a flow rate of 50 mL/min. Powder APR and SNEDDS formulation thermograms were analyzed with the TRIOS program.

#### 2.9.10. X-ray Powder Diffraction (XRPD) Analysis

XRPD analysis was performed with a Shimadzu LabX XRD-6100 (Kyoto, Japan) diffractometer. The X-ray source Cu anode (λ = 1.54060 Å CuKα) was operated at a voltage of 40 kV and a current of 30 mA. After placing the samples in an aluminum sample holder, they were tested at a diffraction angle range of 2θ of 5–60° at intervals of 0.02°.

### 2.10. APR Powder Particle Size and Distribution

The particle size and distribution of the APR powder used in this study were determined by the Malvern MasterSizer 3000 device (Malvern, UK) Hydro MV wet dispersion unit operating based on laser diffraction. The dispersion medium was prepared by adding 25 mg of Tween^®^ 80 (0.06% *w*/*v*) into 40 mL of distilled water. A total of 50 mg of APR powder was added to this dispersion medium. The stirrer speed of the dispersion unit was set to 2400 rpm, and the sonicator power was set to 80% and kept open for 10 min. At the end of the time interval, the sonicator was turned off, and the measurement was taken after 30 s. Measurement results are expressed as Dv(10), Dv(50), Dv(90), standard deviation, and Span value.

### 2.11. In Vitro Lipolysis Test

The in vitro lipolysis study was performed according to the method reported by Zangenberg et al. [34,35] with some modifications. SNEDDS formulation was added to a lipolysis medium containing tris-maleate (2 mM), calcium chloride dihydrate (1.4 mM), sodium chloride (50 mM), bile salt (2.95 mM), and phospholipid (0.26 mM). A 15 min dispersion step was employed in a thermostatically controlled (37 °C) vessel to facilitate the complete dispersion of the SNEDDS formulation. Freshly prepared pancreatic lipase extract was added to the vessel, and lipid digestion was initiated. Samples were withdrawn at the 10th and 15th min of the dispersion phase and at 5, 15, 30, 45, and 60th min of digestion phase. The pH of the medium was adjusted to 6.5 with 0.4 M NaOH solution after sampling. The amount of 0.4 M NaOH solution used to bring the medium back to pH 6.5 was recorded. After 1 mL, samples were withdrawn, and the enzyme activity was inhibited by the addition of 4-bromobenzeneboronic acid (5 µL) solution. Samples were centrifuged at 12,225× *g* for 15 min. The APR amount in the supernatant was measured by validated HPLC method.

### 2.12. In Vitro Dissolution Test

In vitro dissolution tests were performed according to the USP 39—NF 34 monograph and FDA recommendations for APR powder, SNEDDS, and marketed product of APR (Emend^®^). Dissolution studies were carried out using the dissolution tester (Pharmatest, Hainburg, Germany) with paddle apparatus at 100 rpm and 37 °C. Formulations equivalent to 40 mg of APR were added to each vessel. A 2.2% aqueous sodium lauryl sulfate solution (monograph medium) and various buffer solutions (pH 1.2 HCl buffer, pH 4.5 acetate buffer, and pH 6.8 phosphate buffer) were used as the dissolution media. Samples (1 mL) were withdrawn at predefined time points (1, 3, 5, 10, 15, 20, 30, 45, and 60 min) and passed through a 0.45 µm filter. Then, the dissolution volume of the medium was made to reach the initial volume (900 mL) with fresh medium to maintain a constant volume. APR concentration in filtrates was determined by the HPLC.

### 2.13. In Vitro Cytotoxicity Test and MTT Assay

The Caco-2 (ATCC^®^ HTB-37TM) cell line (American Type Culture Collection, VA, USA) derived from human colorectal adenocarcinoma was used in cytotoxicity studies. Cells were maintained in Eagle’s Minimum Essential Medium (Biosciences, DunLaoghaire, Ireland) supplemented with 5% fetal bovine serum (Sigma-Aldrich), 100 U/mL penicillin (Sigma-Aldrich), 100 μg/mL streptomycin (Sigma-Aldrich, Darmstadt, Germany). Cells were grown in an incubator used for cell culture (Thermo Scientific, Dreieich, Germany) at 37 °C, 5% CO_2_. After attaining 80–85% confluence, the cells were harvested.

Firstly, the median lethal dose (LD_50_) of APR was determined. A total of 100 µL of Caco-2 cells at a concentration of 4 × 10^4^ cells/mL in culture medium were seeded in a 96-well plate. Cells were treated with different concentrations (1–100 μM) of APR and then incubated for 24 h. Following this step, 30 µL MTT (3-(4,5-dimethylthiazol-2-yl)-2,5-diphenyltetrazolium bromide) solution (5 mg/mL) was added per well and waited for 4 h. DMSO was added to dissolve the purple formazan crystals formed. The absorbance was measured at 600/570 nm (excitation/emission) using a microplate reader (BMG Optima, Ortenberg, Germany). Triton X-100 was used as a positive control.

Secondly, cells were treated with 20, 30, 50, and 70 µM of APR-loaded SNEDDS formulations. The MTT test was performed as described above.

The percentages of cell viability values were calculated based on the absorbance in non-treated cells (assuming 100% viability). The LD_50_ value of APR was calculated using nonlinear regression with the GraphPad Prism 8. Statistical comparison of pure APR and APR-loaded SNEDDS was performed with one-way ANOVA via GraphPad Prism 8. Differences were considered statistically significant if *p* < 0.05.

### 2.14. Long-Term and Accelerated Stability Studies

Long-term and accelerated stability studies of the gelatin capsule-filled SNEDDS formulation was performed according to the ICH Q1A(R2) guideline. The long-term stability test was carried out at 25 °C ± 2 °C, 60% ± 5% relative humidity (RH), and accelerated stability tests were carried out at 40 °C ± 2 °C, 75% ± 5% RH. In vitro dissolution tests were performed on the samples taken at 0, 1, 2, 3, and 6 months. The changes in the dissolution properties of the formulation during the stability test were examined.

## 3. Results and Discussions

### 3.1. Quantification of APR

Linearity of the calibration curve showed excellent linearity over the concentration range of 1–20 µg/mL. The r^2^ value for the standard curve was above 0.999. Intra-day and inter-day variations of RSD were below 2%, and mean recovery was between 98% and 102%. The limits of detection and quantification values were established at 0.3 and 0.8 µg/mL, respectively. The retention time of drug was about 7.3 min. The method meets the guideline validation requirements.

### 3.2. Excipient Selection for SNEDDS Formulations

Solubilization of the API in the excipients is one of the key aspects of developing a successful SNEDDS formulation. The oily phase and cosolvent selections were based on the results of our previous study [24]. Imwitor^®^ 988 and Capryol^TM^ 90 were selected as the oily phase, and Transcutol^®^ P was selected as cosolvent due to their ability to dissolve more APR. Imwitor^®^ 988 (glycerol monocaprylate, type I) consists of a glycerol ester of caprylic (>90%) and capric (<10%) acid. Capryol^TM^ 90 (propylene glycol caprylate, type II) is an ester (mainly composed of monoesters) obtained by the esterification of caprylic acid with propylene glycol [36,37]. It was concluded that the solubility of APR is higher in oily compounds that are relatively polar and have surface-active properties. The monoglycerides in Imwitor^®^ 988 are compounds with high polarity. In addition, mono- and diglycerides also have surface-active properties [38]. Capryol^TM^ 90 is also a surface-active excipient. It is a nonionic water-insoluble surfactant with an HLB of 5, mainly used as a cosurfactant in oral lipid-based formulations. Transcutol^®^ P dissolved more APR than other investigated cosolvents. It is a well-known excipient known to dissolve many drugs with poor water solubility. Consequently, formulations with the mentioned excipients were developed due to their ability to solubilize APR.

### 3.3. Screening of Surfactants

Kolliphor^®^ CS20, Cremophor^®^ A25, Gelucire^®^ 48/16, Gelucire^®^ 44/14, and Kolliphor^®^ P188 were investigated as solid surfactants. These surfactants were evaluated for emulsification efficiency. The % transmittance values are presented in Table 1.

High % transmittance values in Table 1 indicated that the oil droplets were nano-sized [39]. Kolliphor^®^ CS20 was selected as a surfactant owing to its nanoemulsification superiority with both oily phases. Kolliphor^®^ CS20 is a nonionic surfactant in the structure of polyoxyl 20 cetostearyl ether, with an HLB value of 15 [40]. Using surfactants with high melting points is an innovative method for converting liquid SNEDDS formulation into a solid dosage form. There are various articles in the literature regarding this. Solid surfactants such as Cithrol^®^ DPHS [41], Gelucire^®^ 44/14 [42,43], Gelucire^®^ 48/16 [44,45], and poloxamer 188 [46] were used for this purpose. Problems such as leakage can occur when filling liquid SNEDDS formulations directly into capsules [47]. It is aimed to solidify the formulation inside the capsule body and prevent leakage by using surfactants with a high melting point. The melting point of Kolliphor^®^ CS20 is between 39 and 41 °C, which is high enough for the formulation to solidify in the capsule shell at room temperature yet low enough to melt and disperse rapidly at body temperature.

### 3.4. Equilibrium Solubility of APR in Surfactant Solution

The equilibrium solubility of APR in a surfactant solution is correlated to the maximum amount of APR that can be dissolved in relevant aqueous media. SNEDDS formulations are diluted by gastrointestinal fluids after oral administration. This study offers an opinion about the amount of APR that remains dissolved after dilution. The 1% (*w*/*v*) surfactant solutions correspond to a 100-fold dilution of formulations. The equilibrium solubility of APR in surfactant solutions is shown in Figure 1. Kolliphor^®^ CS20 maintains a higher amount of APR dissolved in the aqueous phase.

### 3.5. Identification of Self-Nanoemulsifying Mixtures

The miscibility and self-nanoemulsifying ratios of the excipients in the ternary systems were investigated. A maximum content of 30% oily phase (Imwitor^®^ 988 or Capryol^TM^ 90) was selected due to self-emulsification considerations. Above this level, it was observed that self-emulsification became difficult. This phenomenon is also compatible with general literature data. The oily phase ratio in SNEDDS formulations is mostly below 30%. Ternary diagrams where the points represent the prepared mixtures investigated are shown in Figure 2. Droplet size measurements of dispersions were plotted using the TriDraw version 2.6 program. Formulations with Imwitor^®^ 988 and Capryol^TM^ 90 are shown in the phase diagrams on the left and right sides of Figure 2, respectively.

The droplet sizes of the formulations ranged between 10 and 376 nm. Formulations with small droplet sizes and unimodal distribution (low PDI value) were preferred due to stability concerns and selected for further evaluation. Selected formulations were coded with the letters of the excipients and their ratios. For example, a formulation consisting of 10% Capryol^TM^ 90, 50% Kolliphor^®^ CS20, and 40% Transcutol^®^ P was coded as CKT 10-50-40.

### 3.6. Short-Term Stability of Drug Loaded Formulations

After formulation selection, 21 mg, 25 mg, 30 mg, and 40 mg APR were dissolved in the SNEDDS preconcentrates. A short-term stability test was performed on APR-loaded formulations. The formulations were dispersed in 100-fold distilled water. The droplet size and distribution of the formed nanoemulsion were measured immediately and 1, 2, and 4 h after dilution. The results are summarized in Table 2. Since droplet size measurements could not be performed due to the rapid precipitation of APR, results for formulations loaded with 40 mg APR are not shown in the table.

It is known that API loading causes changes in droplet size and PDI of dispersions. Droplet size distribution of dispersion can be affected positively or negatively depending on the physicochemical structure of API. For example, if the API accumulates at the oil-water interface and reduces the surface tension, a decrease in droplet size can be observed [48]. On the other hand, incomplete dissolution of the API in formulation or the formation of drug crystals and precipitation over time also affects the droplet size results. Due to the rapid precipitation, DLS measurement was not suitable for 40 mg loaded formulations.

The CKT 20-70-10 formulation could not be loaded with desired amounts of the drug since the solubilization capacity of the formulation was insufficient. Considering the amount of APR that can be loaded (up to 30 mg) and the droplet size stability within 4 h, the most suitable formulation is the CKT 10-50-40 among these seven formulations.

The test duration was chosen based on the pharmacokinetic properties of the APR. Tmax of APR is approximately 4 h. In other words, the absorption is more dominant than the elimination for 4 h after the formulation is taken into the body. Therefore, the formulation in which APR does not precipitate and droplet size distribution does not deteriorate during the absorption phase was preferred.

### 3.7. Increasing the amount of APR Loaded to SNEDDS (Super-SNEDDS)

Adding polymers to SNEDDS formulations has several advantages: Loading API above saturation level (supersaturated SNEDDS), increasing the stability of nanoemulsion droplets after dilution, maintaining a supersaturated state of dispersion for a longer period of time, and slowing down the precipitation rate of API can be considered as examples [49,50]. Due to the fact that nanoemulsion droplets formed from the SNEDDS are thermodynamically unstable, precipitation of the drug over time is expected. Different polymers were used to ensure that the APR remained dissolved for a long time after dilution. Soluplus^®^, Methocel^TM^ E5 (HPMC), Kollidon^®^ VA64, and Kollidon^®^ 25 were tested for this purpose. Supersaturated SNEDDS were prepared as described in Section 2.7. In vitro drug precipitation tests were performed with the formulations loaded with different amounts of APR. Test results are shown in Figure 3.

As seen in Figure 3, polymers slowed the precipitation of the APR in supersaturated formulations loaded with 40 mg of APR. Moreover, Soluplus^®^ completely inhibited the drug precipitation in both concentrations (5% and 10% *w*/*w*). Therefore, 10% Soluplus^®^ was chosen as a polymer additive. Inhibition of drug precipitation could be based on different mechanisms. Reducing the degree of supersaturation due to micelle formation is one of those mechanisms [51]. Soluplus^®^ is a lyophilic polymer and can contribute to droplet stability by forming a multimolecular film around emulsion droplets. Hydrogen bonding (H-bond) may also play a role in preventing precipitation [31]. APR has a carbonyl group (H-bond acceptor) in its structure and forms H-bonds with molecules that are H-bond donors. Soluplus^®^ and Methocel^TM^ E5 have two and one hydroxyl group as H-bond donors, respectively. The reason why precipitation inhibition is more effective with these excipients can be explained by the formation of H-bonds with APR.

### 3.8. Characterization of Optimized SNEDDS Formulation

Optimized formulation contains 100 mg of Capryol^TM^ 90, 500 mg of Kolliphor^®^ CS20, 400 mg of Transcutol^®^ P, 100 mg of Soluplus^®^, and 40 mg of APR. Characterization studies were performed in this formulation. The appearance of optimized solidified SNEDDS formulation is shown in Figure 4.

#### 3.8.1. Nature of the Dispersion Formed

Defining dispersions obtained from SNEDDS formulations as microemulsions is a frequently encountered terminology error in the literature. Microemulsions are thermodynamically stable systems that form spontaneously. However, droplets formed by the dilution of SNEDDS need not be thermodynamically stable [52]. Another difference between microemulsions and nanoemulsions is the mixing order of the ingredients during preparation. In contrast to SNEDDS, the order of mixing of excipients is not crucial in microemulsions.

It was observed that the droplet size and distribution changed significantly (from 9.8 nm to 183.5 nm) when the mixing order of the excipients in the blank formulation was changed. Therefore, it was confirmed that the dispersion formed after dilution is a nanoemulsion.

#### 3.8.2. Results of % Transmittance Measurements

The results of % transmittance measurements in various media are shown in Table 3.

#### 3.8.3. pH Measurement of Dispersions

The pH value of the dispersion obtained by diluting the optimized formulation in water was measured as 4.92 ± 0.02. Measured pH is compatible with gastrointestinal system conditions.

#### 3.8.4. Cloud Point of Formulation

The cloud point of the optimized formulation is higher than the body temperature (>40 °C), thus avoiding phase separation in the gastrointestinal system [53].

#### 3.8.5. Results of Stress Tests

The dispersion obtained by diluting the optimized formulation passed the applied stress tests under forced conditions. Stability problems such as phase separation or drug precipitation were not observed. The absence of visual changes also confirms that the obtained dispersion is nanoemulsion. While microemulsions are sensitive to temperature change, nanoemulsions are highly resistant to changes in ambient conditions.

#### 3.8.6. Results of Emulsification Efficiency and Self-Emulsification Time Tests

The optimized formulation passed the dispersibility test. The formulation dispersed and emulsified within 35 ± 9 s. in the water. The resulting dispersion was nearly transparent with a bluish tinge (Grade B).

#### 3.8.7. Robustness to Dilution

Robustness to dilution test ensures the absence of drug precipitation at higher dilutions in the gastrointestinal system [32]. Since absorption is only possible when the drug is dissolved, drug precipitation may affect in vivo performance. The optimized formulation was exposed to different buffers to simulate the in vivo conditions. Droplet size, distribution, and zeta potential of optimized SNEDDS in various media and dilutions are shown in Table 4.

According to Table 4, it was observed that the droplet size was similar in different media and dilutions. The slightly smaller droplet size in acidic media may be attributed to the higher solubility of APR in acidic environments. Additionally, the droplet size was higher at 1/50 dilutions. However, at the 1/50 ratio, dispersions had a grayish-white appearance, which was undesirable in DLS measurements. Since transparent nanoemulsions were obtained with the 1/100 and above dilution ratios, 1/100 was preferred throughout the study. No significant difference in droplet size was observed at 1/100, 1/500, and 1/1000 dilution ratios. As a result, the optimized formulation was robust to dilution.

Another remarkable point is that the droplet size of the optimized formulation was larger than that of conventional SNEDDS without Soluplus^®^ (Table 2). The fact that Soluplus^®^ has a significant effect on droplet size can be explained by Soluplus^®^ becoming incorporated onto the emulsion droplets (as a hydrophilic colloid) and causing the droplets to grow. In addition, Soluplus^®^ can also form 70–100 nm-sized micelles in aqueous media.

The zeta potential is a significant indicator of the stability of colloids. The stability of the dispersion is considered good when the zeta potential is above +30 mV or below −30 mV [54]. It is observed that the zeta potential values of the dispersions are around 0 mV in different environments. Since the surfactant used in the formulation is nonionic and the other excipients do not have ionizable groups, the zeta potential is expected to be around 0 mV. Considering that SNEDDS formulations emulsify in vivo, long-term colloidal stability is not a cause for concern. Hence, the zeta potential around 0 mV will not pose a problem in terms of stability.

#### 3.8.8. FTIR Spectroscopy

The FTIR spectrum for pure APR, excipients, and optimized formulation are presented in Figure 5.

In the FTIR spectra, peaks were observed in four primary regions except for the fingerprint region. O-H stretching around 3400 cm^−1^ (Capryol^TM^ 90, Transcutol^®^ P, Soluplus^®^), aliphatic C-H stretching between 2850 and 2950 cm^−1^ (all excipients), C=O (ester) stretching around 1700 cm^−1^ (APR, Capryol^TM^ 90, Soluplus^®^) and C=O (amide) stretching around 1600 cm^−1^ (Soluplus^®^) were revealed from spectra. The same peaks mentioned above were also seen in the spectrum of the optimized formulation. The absence of interfering peaks indicates no unwanted interaction between the excipients.

#### 3.8.9. DSC Analysis

The DSC thermograms of APR (Figure 6A), optimized formulation without (Figure 6B) and with (Figure 6C) APR were shown in Figure 6. A sharp endothermic peak at 251.97 °C (normalized enthalpy 77.081 J/g) was observed on the DSC thermogram of the APR. This peak indicates the melting point of crystal-structured APR. The sharpness of the peak suggests that the powder is highly pure. No melting peak was observed in the blank formulation or APR-loaded formulation thermograms. The results suggest that the APR transforms from crystalline to amorphous form. That outcome was also supported by XRPD results. Exothermic events at high temperatures may occur due to the decomposition of excipients.

#### 3.8.10. XRPD Analysis

In the diffractogram of APR powder, sharp peaks were observed in the range of 5–60° at a 2θ angle. Sharp peaks indicated that APR was in crystalline form. It is known that APR has two polymorphs, form I and form II [21]. It is possible to obtain information about the different polymorphs in the powder from the location of the peaks seen in the diffractogram. Polymorphs of APR can be defined between 15° and 25° of the diffractogram. For example, a peak in the range of 20.9°–21.3° (peak maximum at 21.1°) indicates the form II crystals in the mixture. There should not be any peaks in the mentioned range for form I crystals [55]. Characteristic peaks are 15.6°, 17.7°, 22.2° for form I, and 18.3°, 21.1° for form II. According to the diffractogram, it is concluded that the powder used in the study is a mixture of two forms. XRPD diffractograms are shown in Figure 7. The halo pattern in the diffractogram (absence of sharp Bragg peaks) of optimized formulation suggests that the APR is amorphous in the formulation.

### 3.9. APR Powder Particle Size and Distribution

Particle size and distribution of APR powder used throughout the study are presented in Figure 8. When the particle distribution is examined, Dv(10), Dv(50), and Dv(90) values are detected at 7.51 μm, 21.3 μm, and 51.0 μm, respectively. Dv(10), Dv(50), and Dv(90) indicate particle size less than or equal to 10%, 50%, and 90% of sample volume, respectively. Based on these data, the Span parameter of the sample was calculated as 2.042. The volume-based average particle diameter D[4.3] and the specific surface area of the powder sample were calculated by the MasterSizer v3.81 program as 26.6 μm and 254.3 m^2^/kg, respectively.

### 3.10. In Vitro Lipolysis Test

The behavior of the formulations in the medium containing pancreatic lipase was investigated by in vitro lipolysis test. The fate of the formulation in the GI system is more realistically mimicked. Since the optimized formulation does not contain triglycerides, it is classified as a type IV system according to the lipid formulation classification system [56]. After centrifugation, the tubes had two apparent phases. The aqueous phase contains fat acids, bile salts, and drug micelles, and the pellet contains precipitated drugs and calcium soap of fatty acids. The amount of APR dissolved in the aqueous phase and the amount of NaOH consumed to keep the pH at 6.5 are shown in Figure 9. The concentration decrease between 0 and 5 min occurred due to the dilution seen with the addition of pancreatic extract to the lipolysis medium. After 5 min, the solubilization capacity of the optimized formulation remained nearly unchanged. The APR concentration maintained a supersaturated (just above 0.35 mg/mL) state until the end of the lipolysis test. This result could be attributed to the low digestibility of the type IV formulation. The low amount of NaOH consumed during the lipolysis also confirmed that digestion was limited.

### 3.11. In Vitro Dissolution Test

The results of the dissolution test of micronized APR powder (26.6 µm), optimized formulation, and marketed product (Emend^®^) are shown in Figure 10. APR was released considerably faster from the optimized formulation and marketed product. Faster drug release is attributed to nano-sized droplets and particles. Droplets formed from the optimized formulation were around 100 nm, and the particle size of Emend^®^ was reported as approximately 120 nm [57]. The amorphous form of APR also increases the dissolution rate. Because no energy is required to break up the crystal lattices during the dissolution process, the dissolution rate of the amorphous state tends to be higher. The high in vitro dissolution rate generally positively affects in vivo performance of the drug.

### 3.12. In Vitro Cytotoxicity Test and MTT Assay

The toxicity of APR on the Caco-2 cell line was determined, and the result is shown in Figure 11a. According to the nonlinear regression data, the log concentration value corresponding to 50% viability was calculated as 1.669 μM by the GraphPad program. Based on this value, the IC_50_ was determined to be 46.66 μM. The Hill slope of the curve was calculated as −3.059. There was no significant difference (*p* < 0.05) between the formulation without APR and the optimized formulation’s cytotoxicity to the Caco-2 cell line at the tested concentrations (Figure 11b).

The method allows rapid measurement of the activity of dehydrogenases located in mitochondria. It is based on the formation of purple-colored formazan crystals with tetrazolium salts of living and proliferating cells [58]. Ujhelyi et al. [59] performed the cytotoxicity tests of excipients such as Capryol^TM^ 90, Kolliphor^®^ RH40, and Transcutol^®^ P in the HeLa cell line. They reported that the IC_50_ values of these excipients ranged between 0.2 and 5% (*v*/*v*), which was considerably high. Therefore, the excipients used in the optimized formulation can be considered safe. Intravenous and oral LD_50_ values of most nonionic surfactants are above 5 g/kg and 50 g/kg, respectively. Therefore, using around 1–2 g of surfactant can be easily tolerated by the body. Considering that most of the HIV protease inhibitors such as Agenerase^®^, Kaletra^®^, or Norvir^®^ available on the market are prescribed as a few capsules 2–4 times a day, it can be concluded that patients take 2–3 g of Cremophor^®^ derivatives and TPGS [30].

### 3.13. Long-Term and Accelerated Stability Studies

The results of the dissolution test performed on the optimized capsule formulation stored for up to six months in long-term and accelerated stability conditions are shown in Figure 12.

More than 85% (Q + 5%) of the APR was dissolved in the first 30 min at all the time points. The variability was related to the disintegration time of hard gelatin capsule shells. It was visually observed that the capsule shells were opened in the first 10 min of the dissolution test, and the formulation was released into the medium. The formulation in capsules became a liquid state at 37 °C and easily dispersed into the dissolution medium just after the rupture of the capsule shells. Since the dispersion of formulation and dissolution process was rapid, it was observed that a few-minute difference between the rupture time of the capsules shifted the dissolution rate profile. The shift of the dissolution profiles was random and independent of months, which supported the idea that profile shift is related to the capsule disintegration time rather than the change in formulation over time.

## 4. Conclusions

A novel solid SNEDDS formulation containing APR was developed to overcome the problems and shortcomings of the traditional adsorbent-based solid-SNEDDS formulation technologies. The optimized formulation consisted of 100 mg of Capryol^TM^ 90, 500 mg of Kolliphor^®^ CS20, 400 mg of Transcutol^®^ P, 100 mg of Soluplus^®^, and 40 mg of APR, filled into capsules. After the integrity of the capsule shell was disrupted in an aqueous medium, the formulation rapidly dispersed to form droplets sized about 100 nm. Due to the droplet size, optimized formulation showed improved solubility, dissolution profile, and bioavailability over pure drug. APR was released considerably faster from the developed formulation than the marketed product. More than 85% of the APR was dissolved in the first 30 min at physiological pHs. The digestion potential of formulation in the GI system was limited due to the excipients selection. Additionally, the developed formulation becomes prominent with its ease of production, suitability for scale-up, cost-effectiveness, and patient-friendly features. Consequently, an innovative drug formulation that can be an alternative to the existing commercially marketed form has been developed. This drug may be a new option for patients with reduced quality of life due to nausea and vomiting, especially cancer patients.

## Figures and Tables

**Figure 1 pharmaceutics-15-01509-f001:**
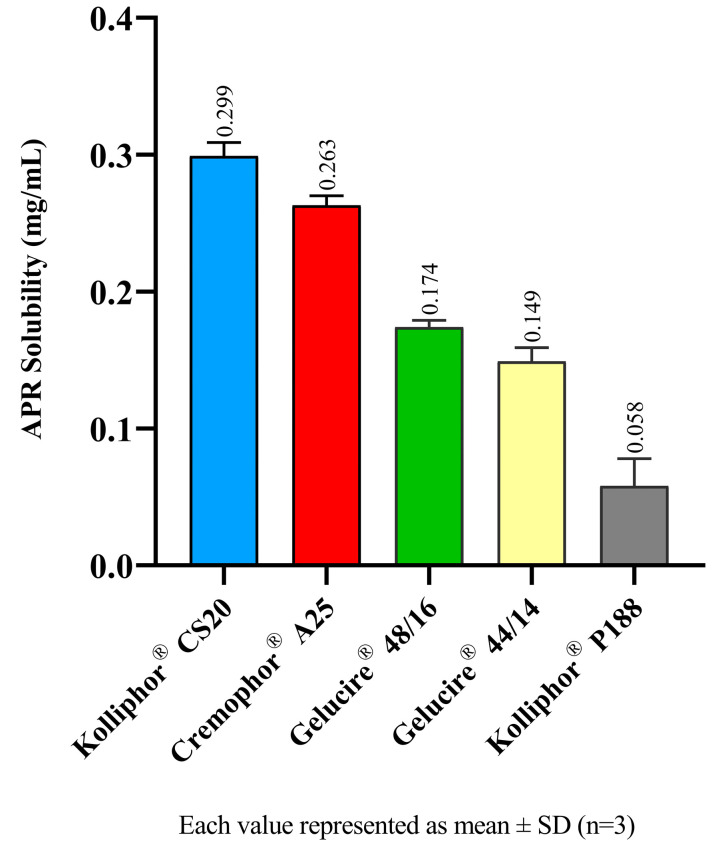
Equilibrium solubility of APR in surfactant solutions.

**Figure 2 pharmaceutics-15-01509-f002:**
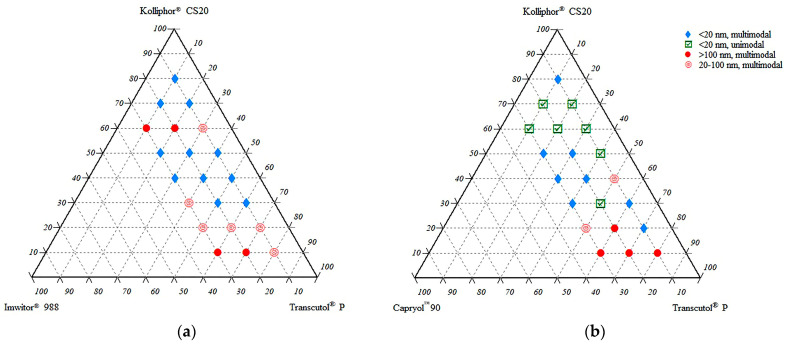
Droplet size and distribution of ternary systems. (**a**) System consisting of Imwitor^®^ 988, Kolliphor^®^ CS20, and Transcutol^®^ P; (**b**) System consisting of Capryol^TM^ 90, Kolliphor^®^ CS20, and Transcutol^®^ P.

**Figure 3 pharmaceutics-15-01509-f003:**
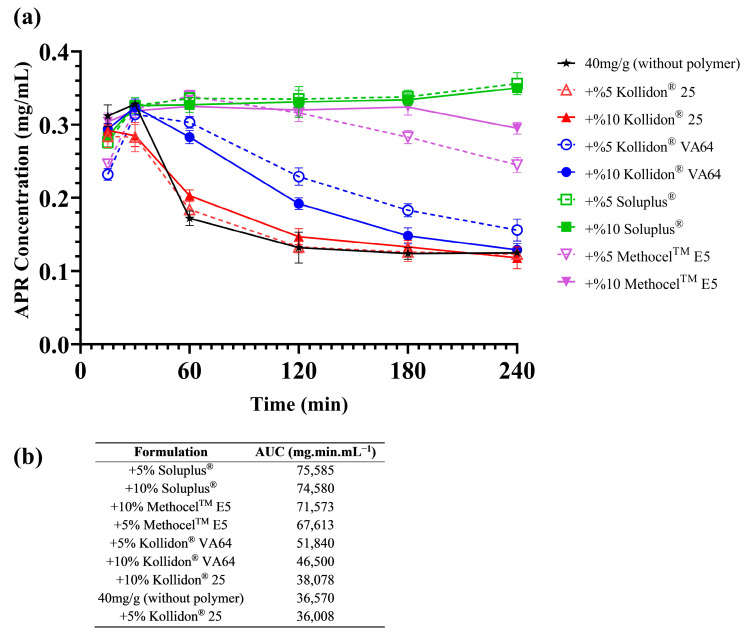
Results of the in vitro drug precipitation test (values represented as the mean of three replicates in the graph): (**a**) Precipitation curves; (**b**) The AUC values of each curve calculated by using averages.

**Figure 4 pharmaceutics-15-01509-f004:**
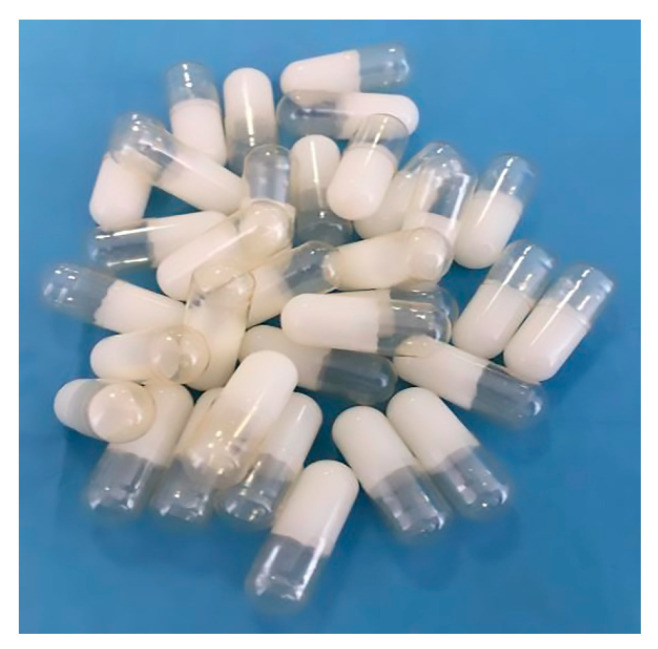
Solidified SNEDDS formulation by filling in hard gelatin capsule 00.

**Figure 5 pharmaceutics-15-01509-f005:**
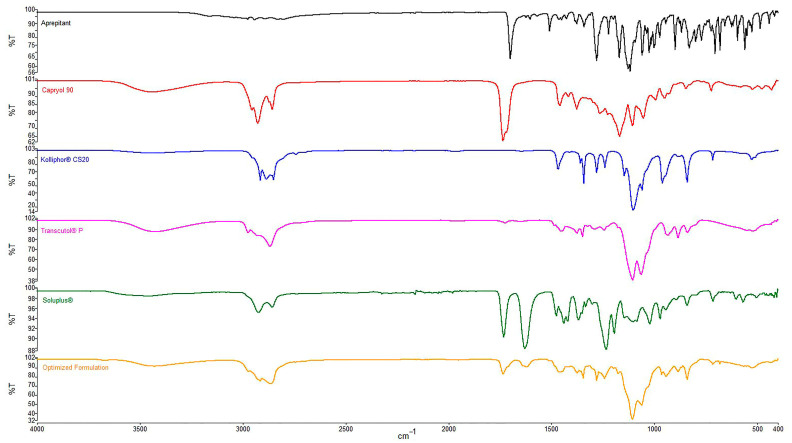
FTIR spectra of APR and excipients.

**Figure 6 pharmaceutics-15-01509-f006:**
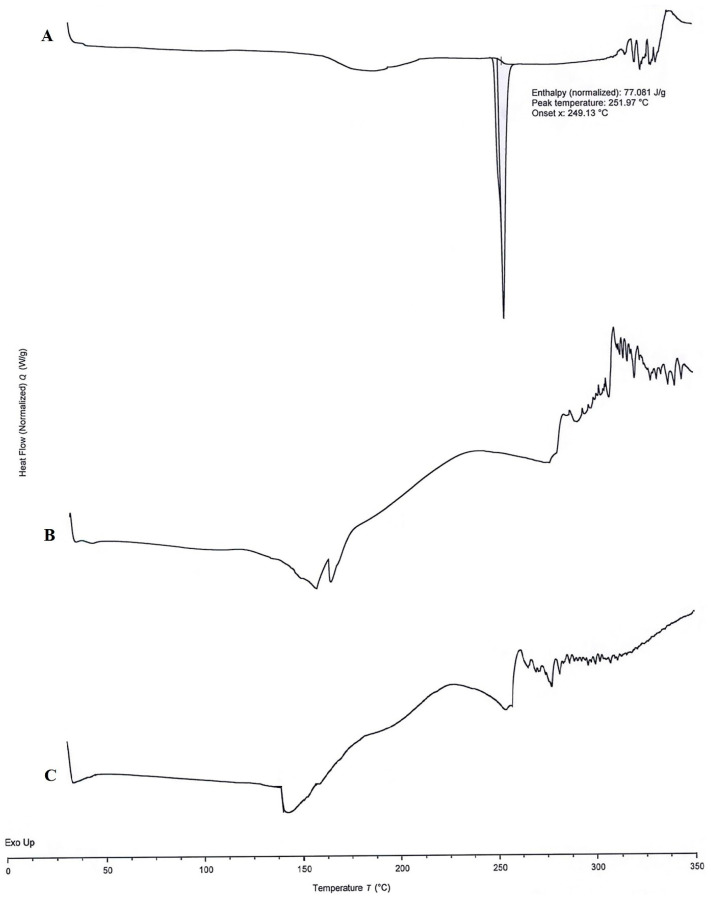
DSC thermograms of APR powder (**A**), blank optimized formulation (**B**), and APR-loaded optimized formulation (**C**).

**Figure 7 pharmaceutics-15-01509-f007:**
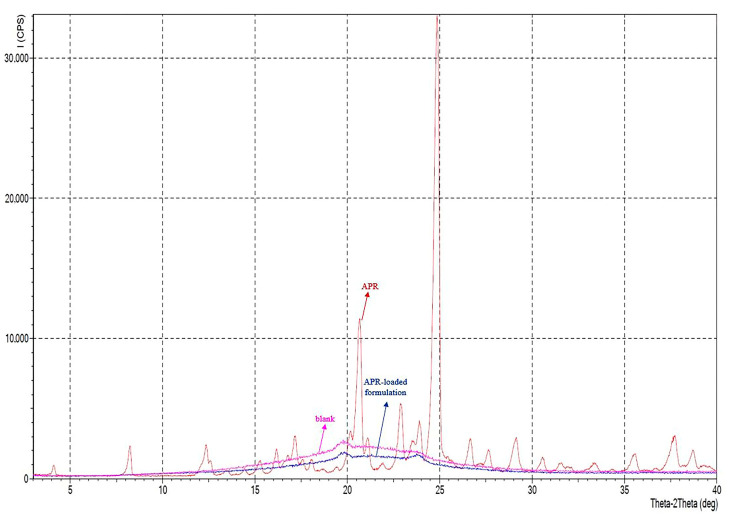
XRPD diffractograms of APR powder (red), blank optimized formulation (pink), and APR-loaded optimized formulation (blue).

**Figure 8 pharmaceutics-15-01509-f008:**
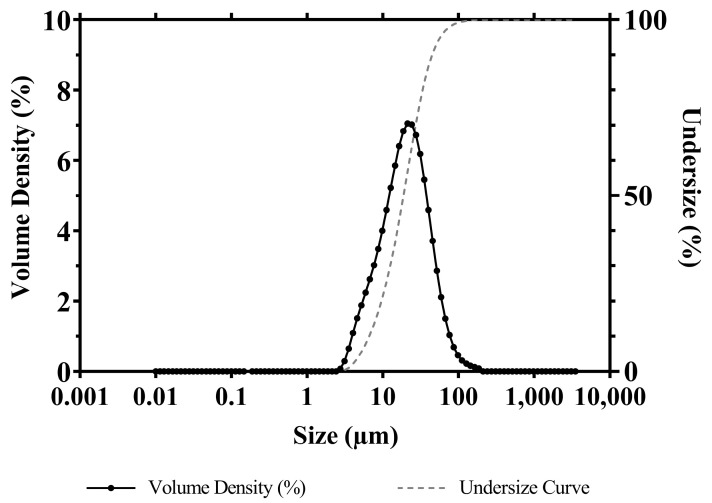
Particle size and distribution of micronized APR powder.

**Figure 9 pharmaceutics-15-01509-f009:**
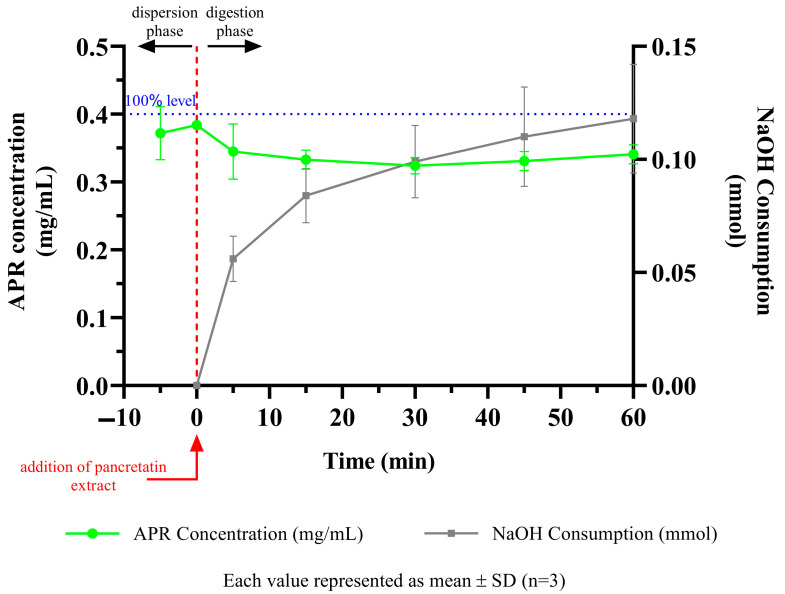
APR concentration and NaOH consumption during digestion of optimized formulation.

**Figure 10 pharmaceutics-15-01509-f010:**
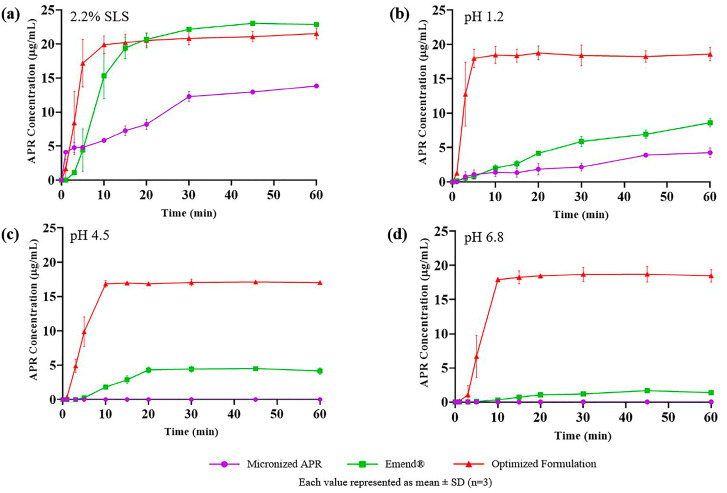
In vitro drug release of micronized APR (purple), Emend^®^ (green), and optimized formulation (red) in (**a**) 2.2% SLS; (**b**) pH 1.2; (**c**) pH 4.5; (**d**) pH 6.8.

**Figure 11 pharmaceutics-15-01509-f011:**
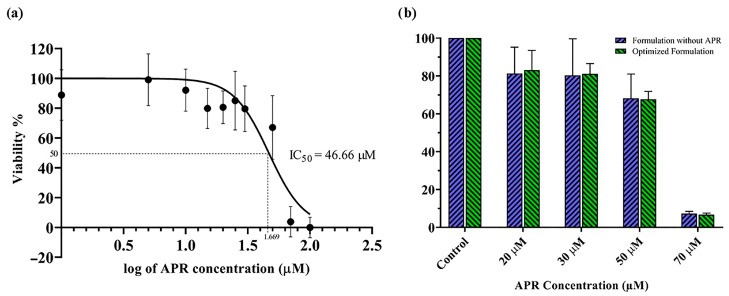
Results of in vitro cytotoxicity tests: (**a**) The toxicity of APR on the Caco-2 cell line; (**b**) % viability results of optimized formulation with and without APR.

**Figure 12 pharmaceutics-15-01509-f012:**
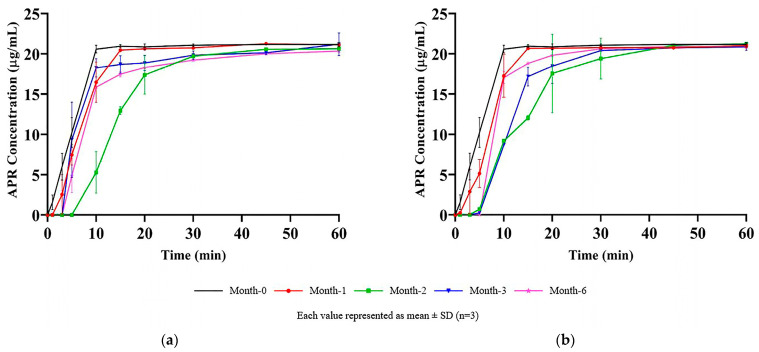
Results of stability studies: (**a**) Long-term stability test; (**b**) Accelerated stability test.

**Table 1 pharmaceutics-15-01509-t001:** Surfactant screening for Imwitor^®^ 988 and Capryol^TM^ 90.

Mixture (1:1)	% Transmittance	Appearance
Imwitor^®^ 988—Kolliphor^®^ CS20	99.55 ± 0.22	Clear
Imwitor^®^ 988—Cremophor^®^ A25	91.58 ± 1.12	Bluish
Imwitor^®^ 988—Gelucire^®^ 48/16	79.69 ± 1.15	Bluish white
Imwitor^®^ 988—Gelucire^®^ 44/14	53.52 ± 1.52	Bluish white
Imwitor^®^ 988—Kolliphor^®^ P188	39.84 ± 3.28	Grayish or dull white
Capryol^TM^ 90—Kolliphor^®^ CS20	96.48 ± 0.34	Clear
Capryol^TM^ 90—Cremophor^®^ A25	93.08 ± 1.27	Bluish
Capryol^TM^ 90—Gelucire^®^ 48/16	86.02 ± 0.72	Bluish
Capryol^TM^ 90—Gelucire^®^ 44/14	57.15 ± 2.02	Bluish white
Capryol^TM^ 90—Kolliphor^®^ P188	77.47 ± 0.52	Bluish white

**Table 2 pharmaceutics-15-01509-t002:** Short-term stability results of selected formulations (size represented as mean ± SD, *n* = 3).

APR Amount		21 mg		25 mg		30 mg	
Formulation Code *	Time	Size (nm)	PDI	Size (nm)	PDI	Size (nm)	PDI
CKT 20-30-50	0 h	17.25 ± 6.36	0.543	188.6 ± 56.9	0.364	278.6 ± 102.3	0.432
	1 h	15.99 ± 4.54	0.322	269.3 ± 76.2	0.320	362.3 ± 84.3	0.359
	2 h	11.26 ± 1.97	0.123	693.5 ± 302.1	0.759	833.5 ± 412.5	0.789
	4 h	11.90 ± 2.86	0.231	1411 ± 705	0.997	1811 ± 906	1
CKT 10-50-40	0 h	10.62 ± 2.18	0.169	11.05 ± 2.36	0.183	10.01 ± 2.07	0.171
	1 h	10.55 ± 2.08	0.155	13.42 ± 3.30	0.242	9.757 ± 1.038	0.045
	2 h	10.02 ± 1.60	0.103	10.21 ± 1.90	0.139	42.87 ± 8.16	0.145
	4 h	10.13 ± 2.00	0.156	12.36 ± 2.17	0.135	872.2 ± 424.6	0.948
CKT 10-60-30	0 h	9.547 ± 1.815	0.145	36.57 ± 5.23	0.082	not dissolved	
	1 h	10.16 ± 1.87	0.136	9.855 ± 1.569	0.101		
	2 h	9.991 ± 1.623	0.106	10.30 ± 1.85	0.128		
	4 h	10.30 ± 2.10	0.166	12.88 ± 2.61	0.222		
CKT 20-60-20	0 h	11.66 ± 3.23	0.306	not dissolved		not dissolved	
	1 h	29.23 ± 6.39	0.191				
	2 h	10.00 ± 2.07	0.171				
	4 h	9.950 ± 2.306	0.215				
CKT 10-70-20	0 h	135.1 ± 31.6	0.218	not dissolved		not dissolved	
	1 h	241.3 ± 73.1	0.367				
	2 h	238.8 ± 71.9	0.362				
	4 h	125.5 ± 32.1	0.261				
CKT 30-60-10	0 h	11.15 ± 2.95	0.281	not dissolved		not dissolved	
	1 h	10.65 ± 2.20	0.171				
	2 h	10.61 ± 2.36	0.198				
	4 h	12.56 ± 3.54	0.318				
CKT 20-70-10	0 h	not dissolved		not dissolved		not dissolved	
	1 h						
	2 h						
	4 h						

* A formulation consisting of 20% Capryol^TM^ 90, 30% Kolliphor^®^ CS20, and 50% Transcutol^®^ P was coded as CKT 20-30-50.

**Table 3 pharmaceutics-15-01509-t003:** Results of % transmittance measurements (*n* = 3).

Water	pH 1.2	pH 4.5	pH 6.8
98.93 ± 0.23	99.32 ± 0.38	96.75 ± 0.52	94.21 ± 1.13

**Table 4 pharmaceutics-15-01509-t004:** Results of Robustness to Dilution (size represented as mean ± SD, *n* = 3).

Media	Dilution	Size (nm)	PDI	ζ Potential (mV)
Water	1/50	104.9 ± 24.53	0.176	−1.57
	1/100	87.10 ± 15.85	0.157	−1.60
	1/500	81.31 ± 23.47	0.237	−4.10
	1/1000	86.17 ± 31.29	0.265	−3.46
pH 1.2	1/50	70.67 ± 12.31	0.145	3.06
	1/100	64.96 ± 12.42	0.149	4.90
	1/500	66.42 ± 15.42	0.205	5.39
	1/1000	69.74 ± 17.23	0.200	6.55
pH 4.5	1/50	106.5 ± 19.72	0.135	−0.14
	1/100	83.43 ± 15.18	0.140	−0.48
	1/500	92.02 ± 18.77	0.161	−0.69
	1/1000	78.34 ± 19.94	0.205	−0.38
pH 6.8	1/50	156.9 ± 42.59	0.212	0.22
	1/100	98.89 ± 18.04	0.140	−1.54
	1/500	90.43 ± 15.94	0.136	−1.93
	1/1000	101.6 ± 30.71	0.236	−3.89

## Data Availability

All data is contained within this article.

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
