# Peer review of "A Novel Semi-Solid Self-Emulsifying Formulation of Aprepitant for Oral Delivery: An In Vitro Evaluation"

_pharmaceutics, 2023, doi:10.3390/pharmaceutics15051509_

Round 1

Reviewer 1 Report

The research work “A Novel Semi-Solid Self-Emulsifying Formulation of Aprepitant for Oral Delivery: An In Vitro Evaluation” is studying a self-emulsifying formulation that can be filled into capsules in a melted state and then solidified at room temperature. The proposed work is targeting to practice solutions compare to marketed nanocrystals. The excipient used is CapryolTM 90, Kolliphor® CS20, Transcutol® P, and Soluplus®. The formulation was characterized and optimized based on DLS, FTIR, DSC, and XRPD techniques, as well a lipolysis test was conducted to predict the digestion performance of formulations in the gastrointestinal system. The formulation also studied drug release and cytotoxicity in the Caco-2 cell line to prove, improve solubility and reduce toxicity. The observation and comments to improve the manuscript are as follows:

1.       Line 148- 152, 2.6. Short-term Stability of Drug Loaded Formulations APR was loaded on the formulations that passed the preliminary quality test based on droplet size. Short-term stabilities of drug-loaded formulations (20–40 mg) were examined. Measurements were repeated at 0, 1, 2, and 4 h, and changes in the droplet size and PDI were analyzed to evaluate the short-term stability of the formed nanoemulsions.

Is this the correct way to find short-term stability, just 4 h study?

2.       As per 2.7 methods, why heating is done twice? Will second-time heating to dissolve the drug will not affect SEEDS, stability?

3.       2.14. Long-term and Accelerated Stability Studies, why only changes in the dissolution 309 properties, were examined? Globule size and zeta is also important characteristics as far as permeation and bioavailability is concerned.

4.       Table 1: % Transmittance, if its avg value please add Standard Deviation

5.       Table 1 composition. Why was only a 1:1 ratio tested? Not 1:2 and 2:1?

6.       Line 415, Super-SNEDDS? Why use this word for any specific reason?

7.       Figure 3: Clarity of formulation AUC..figure 3b is not there. Use a) and b) for providing separate descriptions.

8.       Table 3. Results of % transmittance measurements (n=3), Standard deviation?

9.       Why pH is s 4.92 ± 0.02? Which excipient imparts to it. Is the drug table at this pH

10.   3.8.5. Results of Stress Tests, done via visual observation?

11.   Line 482-483: The formulation dispersed and emulsified within 1 minute in the water. Please provide an accurate time.

12.   Conclusion: How the author can conclude, Line 649-651: the developed formulation be- 649 comes prominent with its ease of production, suitability for scale-up, cost-effectiveness, 650 and patient-friendly features

Author Response

The authors appreciate the reviewers for evaluating the manuscript and giving contributions. Manuscript changes can be tracked by using the “Track Changes” function of MS Word.

Reviewer 2 Report

Herein, authors developed semi-solid self-emulsifying formulation of aprepitant (APR) for oral delivery. They screened surfactants and polymers and optimized the formulations by different solidification approaches. As the result, the obtained formulations achieved high drug loading efficiency, improved solubility, and low toxicity, providing a potential option for patients suffered from CINV. Overall, the manuscript was well organized. Thus, I suggest its acceptance after addressing following issues.

1.     I suggest section 3.1 should be integrated into the Methods section.

2.     In line 87, what did API represent? Apparently, the API and APR (abbreviated for aprepitant) might confuse the readers.

3.      Trademarks were usually not presented in the manuscript.

4.     In table 1, Why did authors choose the 1:1 for the surfactant screening. Moreover, was it weight ratio, volume ratio, molar ratio, or others?

5.     In Figure 6, images with higher resolution should be provided since the names of different groups were not clear.

6.     I strongly suggest that the figures presented should be unified throughout the manuscript.

7.     The biological effect of released APR should be evaluated.

8.     Although authors claimed that the current study is an in vitro evaluation, the in vivo study such as pharmacokinetics must be conducted; otherwise, the in vitro results are not convincing enough to support their conclusion.

The quality of English is fine. Meanwhile, it should be carefully checked and revised to avoid the spelling, expression and grammar errors. 

Author Response

(The authors gave the same response as above.)

Reviewer 3 Report

This paper is a well-written report with sufficient novelty.

Would like to ask the authors if the in vitro dissolution test was made according to the Pharmacopeia? If such, please state it in the methods.

Figure 5 is not needed, as particle size and PDI are already disclosed by particle size and PDI values that refer to the particle size distribution. It’s redundant information.

Figure 11 caption should be more detailed, particularly regarding the composition of the different media as well as the samples.

Standard deviation should be added in all graphs.

Conclusions must be improved as they do not add any relevant information as they are presented now. Quantitative information should be added.

Also, authors should make a graphical abstract.

Author Response

(The authors gave the same response as above.)
